# The population genetics of the causative agent of snake fungal disease indicate recent introductions to the USA

Jason T. Ladner [1] *, Jonathan M. Palmer[2]¤, Cassandra L. Ettinger[3], Jason E. Stajich[3], Terence M. Farrell[4], Brad M. Glorioso[5], Becki Lawson[6], Steven J. Price[7], Anne G. Stengle[8], Daniel A. Grear[9], Jeffrey M. Lorch [9]*

1 Northern Arizona University, Flagstaff, Arizona, United States of America, 2 Independent Researcher, Palo Alto, California, United States of America, 3 University of California-Riverside, Department of Microbiology and Plant Pathology, Riverside, California, United States of America, 4 Stetson University, Department of Biology, DeLand, Florida, United States of America, 5 U.S. Geological Survey—Wetland and Aquatic Research Center, Lafayette, Louisiana, United States of America, 6 Institute of Zoology, Zoological Society of London, Regent's Park, London, United Kingdom, 7 University of Kentucky, Department of Forestry and Natural Resources, Lexington, Kentucky, United States of America, 8 Holyoke Community College, Division of Science, Engineering, and Mathematics, Holyoke, Massachusetts, United States of America, 9 U.S. Geological Survey—National Wildlife Health Center, Madison, Wisconsin, United States of America

¤ Current address: Genencor Technology Center, Palo Alto, California, United States of America
* jason.ladner@nau.edu (JTL); jlorch@usgs.gov (JML)

**Data Availability Statement:** Metadata associated with this project are available at https://doi.org/10.5066/P9J2MCLJ. Additional code and output for various analyses have been deposited in OSF and

## Abstract

Snake fungal disease (SFD; ophidiomycosis), caused by the pathogen *Ophidiomyces ophiodiicola* (*Oo*), has been documented in wild snakes in North America and Eurasia, and is considered an emerging disease in the eastern United States of America. However, a lack of historical disease data has made it challenging to determine whether *Oo* is a recent arrival to the USA or whether SFD emergence is due to other factors. Here, we examined the genomes of 82 *Oo* strains to determine the pathogen's history in the eastern USA. *Oo* strains from the USA formed a clade (Clade II) distinct from European strains (Clade I), and molecular dating indicated that these clades diverged too recently (approximately 2,000 years ago) for transcontinental dispersal of *Oo* to have occurred via natural snake movements across Beringia. A lack of nonrecombinant intermediates between clonal lineages in Clade II indicates that *Oo* has actually been introduced multiple times to North America from an unsampled source population, and molecular dating indicates that several of these introductions occurred within the last few hundred years. Molecular dating also indicated that the most common Clade II clonal lineages have expanded recently in the USA, with time of most recent common ancestor mean estimates ranging from 1985 to 2007 CE. The presence of Clade II in captive snakes worldwide demonstrates a potential mechanism of introduction and highlights that additional incursions are likely unless action is taken to reduce the risk of pathogen translocation and spillover into wild snake populations.

are available at https://osf.io/fmbh5/. Sequence data, genome assemblies, and genome annotations associated with this project were deposited in GenBank under BioProject PRJNA780910.

**Funding:** The authors received no specific funding for this work.

**Competing interests:** The authors have declared that no competing interests exist.

**Abbreviations:** AAFTF, automatic assembly for the fungi; BUSCO, Benchmarking Universal Single-Copy Orthologs; EID, emerging infectious disease; HMG, high mobility group; HPD, highest posterior density; Oo, *Ophidiomyces ophiodiicola*; PCA, principal component analysis; SFD, snake fungal disease; SNP, single nucleotide polymorphism; tMRCA, time of the most recent common ancestor.

## Introduction

Emerging infectious diseases (EIDs) are increasingly recognized as having major impacts on wildlife conservation and global biodiversity [1–3]. Anthropogenic activities resulting in movement of pathogens to new locations and environmental change are the driving forces behind the emergence of many wildlife diseases [1,2,4–6]. Due to several characteristics that make them pernicious pathogens (e.g., high virulence, ability to establish environmental reservoirs, and broad host range), fungi have been responsible for some of the most consequential of these EIDs [7]. For example, the emergence of white-nose syndrome, a disease of hibernating bats caused by *Pseudogymnoascus destructans*, has caused population declines in excess of 90% for several bat species across eastern North America [8]. Similarly, *Batrachochytrium dendrobatidis*, the causative agent of chytridiomycosis, has been linked to the global decline of hundreds of amphibian species and the extinction of at least 90 species [9].

Mechanisms for disease emergence are complex, but typically fall somewhere along a gradient between 2 broad categories: (1) introduction of an exotic pathogen into a naïve host population ("novel pathogen hypothesis") or (2) in situ emergence of a native pathogen due to changes in environmental, host, or pathogen characteristics that alter disease ecology ("endemic pathogen hypothesis") [10]. Although strict adherence to these binary categories may be an oversimplification in some cases (disease emergence mechanisms are often multifactorial), understanding whether a pathogen is native or introduced to a region is a key first step for assessing the threat of a disease to host populations and predicting pathogen spread, as well as for devising disease management strategies. For example, while targeted containment or eradication efforts may be effective against novel pathogens, successful management of endemic pathogens is more likely to involve manipulation of factors (e.g., environmental, host) that contribute to disease outbreaks [10].

Despite the importance of identifying which broad category best explains the reason for emergence of a disease, the origin of fungal pathogens is often elusive, and the sources of many important fungal (and fungal-like) pathogens remain unknown (e.g., *Bretziella fagacearum*, causative agent of oak wilt [11]; *Ophiognomonia clavigignenti-juglandacearum*, causative agent of butternut canker [12]; *Phytophthora ramorum*, causative agent of sudden oak death [13]). A lack of historical samples and surveillance, temporal delays between actual emergence and recognition of the disease, and the potential for multiple introductions and recombination between strains can make it difficult to ascertain the extent of a fungus' geographic distribution and whether it is native or introduced to a region [14–16].

Snake fungal disease (SFD or ophidiomycosis) is often cited as a fungal EID afflicting wildlife [17,18], although the extent to which SFD is actively emerging is difficult to quantitatively assess [19,20]. The disease first gained attention in 2008 when severe fungal infections manifested in a well-studied population of eastern massasauga rattlesnakes (*Sistrurus catenatus*) in Illinois, United States of America [21]. Subsequent investigations revealed that SFD was widely distributed in the eastern USA and Great Lakes region of Canada [18,20,22]. The fungus *Ophidiomyces ophiodiicola* (*Oo*) is the causative agent of SFD [23,24]. Although *Oo* is not known to infect other animals, the fungus has a broad host range among snakes and most snake taxa are predicted to be susceptible [25]. However, disease severity and outcomes of infection are variable [18], and the broader conservation impacts of SFD are difficult to predict, in part because the extent of, and mechanism behind, the disease's emergence within snake populations is unknown.

SFD is now endemic in wild snakes in eastern North America [18,20]. However, existing evidence as to the origin of *Oo* in the eastern USA is ambiguous. Under the novel pathogen hypothesis, we would predict that *Oo* arrived in the USA relatively recently, perhaps in the

order of decades prior to the increase in cases of SFD reported during the early 2000s [18,21]. Indeed, the detection of genetically unique and diverse strains of *Oo* from wild snakes in Eurasia [26,27] is consistent with this hypothesis, as it indicates that *Oo* may have originated from outside North America. In contrast, the endemic pathogen hypothesis would be consistent with a much earlier arrival of *Oo* in the USA, with the recent increase in SFD cases linked to greater awareness or environmental changes rather than a new introduction. *Oo* is thought to be a specialized pathogen of snakes with limited ability to survive as an environmental saprobe [28]. Thus, under the endemic pathogen hypothesis *Oo* most likely evolved in North America or arrived in the USA through the natural migration of snakes from Eurasia to North America via Beringia, which occurred in the span of 27 to 55 million years ago [29,30]. Detections of SFD in snakes in the USA as early as 1945 [31] and the lack of a documented wave-like spread pattern after the reported emergence of the disease in the early 2000s [18] could support the endemic pathogen hypothesis. However, a lack of sufficient historical data on the disease has made it difficult to trace the origins of *Oo* in North America.

Phylogenetic and population genetic analyses based on whole-genome sequence data provide powerful tools for inferring the origins of fungal pathogens in the absence of historical samples [32]. For example, Drees and colleagues [33] demonstrated that isolates of *P. destructans* from North America exhibited minimal genetic diversity, consistent with clonal expansion following a single introduction event from Eurasia. In the case of *B. dendrobatidis*, analyses based on whole-genome sequence data ultimately revealed Southeast Asia as the likely source of the fungus—a finding that was long evasive due to multiple introduction events involving several genetic lineages and recombination of those lineages upon recontact [15,16]. Here, we report full genome sequences of 82 strains of *Oo* and phylogenetic and population genetic analyses that explore the origins of *Oo* in North America. Our findings indicate that strains of *Oo* in the eastern USA are primarily represented by 4 clonally expanded lineages or hybrids between those lineages, and that the ancestors of these clonal lineages arrived in the region relatively recently.

## Results

### Whole-genome sequencing and annotation

Using the Illumina platform, we obtained whole-genome shotgun sequencing data for 82 strains of *Oo*, with 1.2 to 10.8 M paired end reads obtained for each (median = 2.4 M). After quality filtering and trimming, this provided 9.3 to 202.4× average genome coverage depth (median = 23.5×; S1 Table). Using these data, we assembled both the nuclear and mitochondrial genomes de novo, with 81 to 1,350 nuclear contigs per strain (median = 411.5) and a single mitochondrial contig per strain (length = 42,585 to 54,364 bp). We used the de novo contigs to predict open reading frames and protein sequences (6,901 to 7,190 predicted open reading frames per strain). A Benchmarking Universal Single-Copy Orthologs (BUSCO) analysis with 4,862 genes estimated that these assemblies were 91.1% to 93.4% complete (median = 92.9%), which is on par with the only *Oo* genome assembly currently present in NCBI (GenBank: MWKM00000000.1, 91.9% complete according to BUSCO) [34].

### Mating type

Predicted mating type proteins identified in *Oo* shared 43.6% to 61.7% amino acid identity with the conserved portion of the alpha box mating type protein MAT1-1 and 44.4 to 62.7% amino acid identity with the conserved portion of the high mobility group (HMG) domain-containing MAT1-2 protein of other Onygenalean fungi. Each *Oo* strain possessed either a putative *MAT1-1* or *MAT1-2* locus, consistent with a heterothallic mating system. Both mating

type idiomorphs were observed in isolates from wild snakes in the USA and in captive snakes (S2 Table). However, all 4 European isolates sampled possessed the *MAT1-2* idiomorph. To further explore mating type idiomorphs present in Europe, we screened 9 additional strains of *Oo* isolated from wild snakes in the United Kingdom that were not included in the whole-genome sequence analysis for *MAT1-1* and *MAT1-2* using targeted PCR assays (see Supporting Text). Only *MAT1-2* was detected in these 9 strains.

## Phylogenetic analysis

To obtain a genome-wide view of the evolutionary relationships among the 82 *Oo* strains, we generated a maximum-likelihood phylogeny using a concatenated amino acid alignment of 5,811 proteins (Figs 1A and S1A). This represented approximately 81% to 84% of the annotated protein coding genes for each strain, and the overall topology, which consisted of 3, well-supported main clades, was consistent with previously published phylogenies for *Oo* [26,27]. One of these clades (Clade I) contained all 4 strains isolated from wild snakes in Europe; a second clade (Clade II) contained all 65 strains isolated from wild snakes in North America as well as 10 strains from captive snakes, and the third clade (Clade III) included 3 strains, all collected from snakes in captivity. Notably, strains from Clades II and III were observed in captive snakes from all 3 sampled continents (North America, Europe, and Australia). Consistent with previous studies, we found that Clade III was an outgroup to Clades I and II; however, our full genome analysis indicated a much higher level of genetic divergence than previously reported. The 2 protein coding genes used in previous phylogenetic analyses of *Oo* (actin and translation elongation factor 2α [26,27]) exhibited average levels of divergence between Clades III and I/II (0.016% and 0% at the amino acid level) that were lower than approximately 97% to 98% of the orthologs we examined (S1B Fig). Average per gene amino acid divergence between Clades III and I/II was 2.6% (median = 1.9%).

We also observed the same 3 well-supported clades in a phylogeny based on a nucleotide-level single nucleotide polymorphism (SNP) alignment across the full mitochondrial genome (S1C Fig). However, while mitochondrial genetic diversity within each clade was generally low (<0.028%, <0.018%, and ≤0.133% divergence for Clades I to III, respectively), 1 strain from Clade II (NWHC 44736–75) exhibited a high level of genetic divergence compared to the other members of this clade (0.20% to 0.21%). Levels of genetic divergence between strain NWHC 44736–75 and Clades I and III were even higher (0.32% to 0.35% and 1.53% to 1.56%, respectively). Combined with phylogenetic placement based on the nuclear genome, this is consistent with strain NWHC 44736–75 being of hybrid origin between Clade II and a lineage that is not represented in our dataset.

To better resolve relationships within the North American and European clades (i.e., through inclusion of more informative sites), we removed the 3 highly divergent strains from Clade III and performed a nuclear genome-wide phylogenetic analysis at the nucleotide level. This phylogeny was based on a 19,699,744 base pair (bp) alignment, which included 189,199 variable sites. For Clade I, which includes the 4 European strains, we observed strong concordance between the patterns of genetic divergence in the nuclear and mitochondrial genomes (Fig 1). However, no such concordance was observed within Clade II, which includes all of the North American strains.

While we observed several apparently clonal lineages within Clade II, many of the strains from this clade were highly divergent from all other strains. In contrast, just 3 mitochondrial genotypes accounted for 96% of all Clade II strains (72/75), all 3 of these genotypes were observed in strains that exhibited substantial levels of genetic divergence within the nuclear genome, and all 3 exhibited polyphyletic nuclear relationships (Fig 1). Additionally, despite

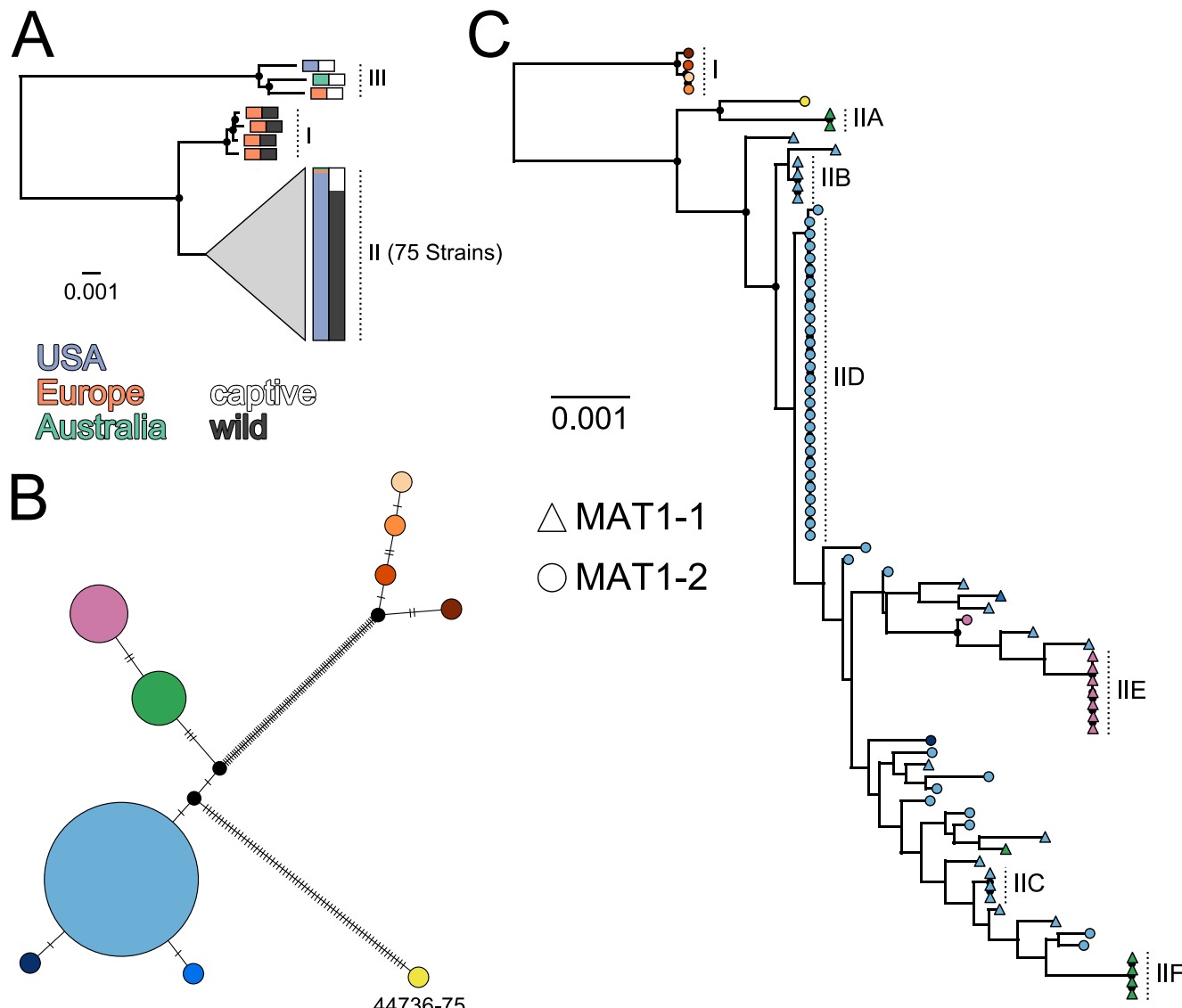

**Fig 1. All North American isolates of *Ophidiomyces ophiodiicola* (*Oo*) form a well-supported clade, but there is discordance between genetic relationships within the nuclear and mitochondrial genomes.** (A) Maximum-likelihood phylogeny including all 82 *Oo* strains and based on an amino acid-level alignment consisting of 5,811 nuclear proteins and 3,311,400 positions. Colors indicate the location from which the strain was obtained (blue, orange, and green) and whether the strain was isolated from a captive (white) or wild (black) snake. Clade II, which contains 75 strains, has been collapsed and the colored bars to the right of this clade indicate the proportion of strains from each category. See S1 Fig for an uncollapsed version of this tree. (B) A median-joining haplotype network including all 79 strains from Clades I and II and built from 159 SNPs present in the mitochondrial genome ("best" SNPs from NASP). Haplotype colors correspond to the tip colors from (C). Each tick mark corresponds to a single SNP difference and black circles represent inferred, but unsampled nodes. (C) Maximum-likelihood phylogeny including all 79 strains from Clades I and II and based on an alignment of 19,699,744 base pairs, including 189,199 SNPs from the nuclear genome. Tip shapes indicate the mating type of each strain, and colors correspond to the mitochondrial genotypes presented in (B). Vertical dotted lines and labels indicate the strains of Clade I, as well as the putative clonal lineages within Clade II. In both (A) and (C), filled black circles indicate nodes with bootstrap support ≥90 and the scale bar represents the horizontal length equivalent to 0.001 changes per site. Data underlying this figure can be found in OSF: https://osf.io/fmbh5/. SNP, single nucleotide polymorphism.

long branch lengths in the nuclear phylogeny, most of the nodes within Clade II exhibited low bootstrap support. These patterns all indicated a history of recombination within Clade II (see below). However, we did not find any evidence of recombination among strains contained within the same putatively clonal lineage. Specifically, the nuclear phylogeny supported 6

apparently clonal lineages (Fig 1, IIA–IIF), each of which included 2 to 27 *Oo* strains. All of the strains within a given clonal lineage shared an identical mitochondrial genotype and had the same mating type locus. Additionally, we found little, if any, evidence for homoplasy within these lineages based on a parsimony analysis with PAUP* (consistency indexes = 0.93–1, only run for the 4 clonal lineages that contained ≥4 strains) (S3 Table). We also saw little evidence of homoplasy among the strains of Clade I (consistency index = 0.99).

## Recombination analyses

We observed a very low consistency index (0.25) in our analysis that included all 75 *Oo* strains from Clade II (S3 Table), which is suggestive of a history of recombination. Specifically, this indicated that the most parsimonious tree for this dataset required 4 times the minimum number of required changes because of the presence of homoplasy. For each of the 17 scaffolds with >50,000 high confidence base calls, we also broadly assessed recombination using the Phi test [35]. We detected widespread evidence of recombination among the *Oo* strains of Clade II, with significant evidence for recombination observed on each of the 17 scaffolds, even after correcting for multiple tests (S2 Fig).

To further explore the genetic relationships among *Oo* strains and to look for evidence of recent recombination within Clade II, we generated a co-ancestry matrix using ChromoPainter [36]. A principal component analysis (PCA) based on our co-ancestry matrix revealed patterns consistent with recent recombination among 3 of the clonal lineages in Clade II: IID, IIE, and IIF (Fig 2B). Specifically, we found that 25 of the *Oo* strains fell along a gradient between IID and IIF, while 8 strains fell along a gradient between IID and IIE, and 1 additional strain (NWHC 24411–1) appeared to fall along one of these gradients, but very close to the IID clonal lineage. These patterns are consistent with recent hybridization between these lineages as all IID strains possessed the *MAT1-2* mating type locus, whereas the strains from IIE and IIF all possessed the *MAT1-1* mating type locus. The putative hybrids contained a mixture of these 2 mating types. Only 2 examples of clonal expansion within a recombinant lineage were observed (IIB and IIC). In both instances, the clonal strains within the recombinant lineages were collected from the same location (Norfolk County, Massachusetts, USA for IIB; Virginia Beach, Virginia, USA for IIC), consistent with limited geographic distributions; however, even with modest sample sizes (3 to 4 strains), each of these clonal lineages was isolated from multiple snake species. All other hybrids were represented by a single isolate and appeared to indicate separate recombination events (Fig 2C).

We also compared these apparent hybrid strains to their putative parental lineages across all of the variable sites in the nuclear genome, and we found that ≥98% of the SNPs from these putative hybrids could be explained by a combination of the variants observed in lineages IID and IIE or IIF. In fact, in 83% of these strains (29/35), >99.5% of the variable sites could be explained by a combination of 2 of these clonal lineages. In contrast, <86% of the variable sites could be similarly explained for the strains from lineage IIA, which did not fall along either of these hybridization gradients. Notably, however, several of the hybrid strains appeared to share ancestry with lineages IID, IIE, and IIF, which indicated multiple episodes of hybridization (Fig 2). Repeated hybridization was also consistent with the high variability among recombinants in estimated ancestry from each clonal lineage, as observed in both the PCA analysis (Fig 2B) and an analysis of fixed SNPs within each clade (Fig 2C).

We also used PAUP* to look for evidence of homoplasy after excluding the hybrid strains. We observed a substantial increase in the consistency index within Clade II after excluding recent recombinants (i.e., only including lineages IIA and IID–F): 0.93 versus 0.25 for all of Clade II (S3 Table). This is consistent with these recent hybrids being responsible for most of

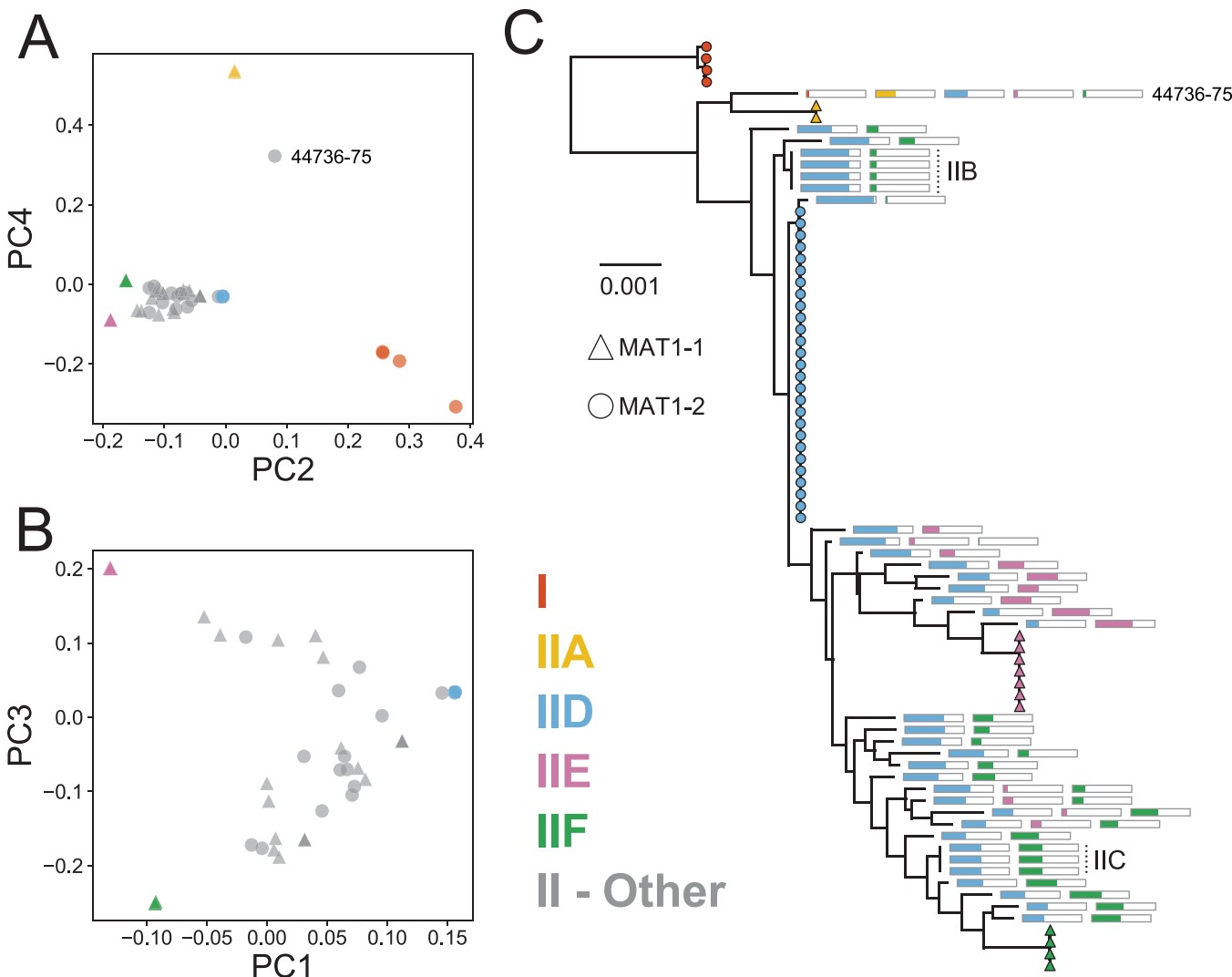

**Fig 2. Many North American strains of *Ophidiomyces ophiodiicola* are recombinants between clonal lineage IID and clonal lineages IIE and/or IIF.** (A) and (B) illustrate the genetic similarity among the strains of Clades I and II based on a co-ancestry matrix generated using ChromoPainter and visualized through PCs 1–4. Panel (A) includes all 79 strains, while the strains of Clade I and clonal lineage IIA, as well as strain NWHC 44736–75, have been excluded from panel (B). Shapes indicate the mating type of each strain. Strains from nonrecombinant clonal lineages are indicated by colored shapes, recombinant strains are shown in gray. (C) Nuclear maximum-likelihood phylogeny (same as shown in Fig 1C) with colored bars approximating the level of ancestry from each of the nonrecombinant clonal lineages. Strains belonging to nonrecombinant clonal lineages are indicated with colored shapes, with the shape indicating mating type. For recombinant strains, each bar represents a set of nuclear SNPs that are specific to 1 of the clonal, nonrecombinant lineages, and the proportion of the bar filled with color indicates the proportion of those SNPs that were also observed in that particular strain. A particular bar is only shown if the recombinant strain's genome contained at least 1 SNP indicative of that clonal lineage. Colors are the same as those used in (A, B). Vertical dotted lines and labels indicate the recombinant clonal lineages. Data underlying this figure can be found in OSF: https://osf.io/fmbh5/. PC, principal component; SNP, single nucleotide polymorphism.

the homoplasy observed within this clade. However, we still observed more homoplasy than we saw within most of the individual clonal lineages, and the consistency index was even lower when we included the 4 strains from Clade I (0.86). Therefore, despite the absence of recent recombination within each clonal lineage, we saw evidence for historical recombination between lineages.

Geographically, clonal lineage IID was the most widespread in the USA, with strains recovered from 14 eastern states (S3 Fig). Although less frequently sampled, clonal lineage IIE was also widespread, while clonal lineage IIF was more geographically restricted, which could be an artifact of small sample size (IIF was represented by only 2 strains from wild snakes). Clonal lineage IIA and a related hybrid strain (NWHC 44736–75) were represented by single strains from wild snakes. Geospatial patterns were evident among hybrid strains as well. Specifically, hybrids between lineages IID and IIE were found in the southeastern region of the USA, while hybrids between lineages IID and IIF were found throughout the Atlantic and Gulf Coastal Plains regions. Hybrids of these clonal lineages were rarely sampled in the Midwest region of the USA.

Within nonrecombinant clonal lineages, we examined the relationship between genetic divergence and geographic distance by comparing pairwise nucleotide-level divergence based on the nuclear genome with the distance between localities from which each wild snake was captured. We observed a strong positive correlation between genetic divergence and distance across relatively short geographic distances, but correlation was lacking between these metrics across larger geographic distances (S4 Fig). This pattern was consistent with the presence of fine-scale population structure and an absence of recent long-distance dispersal.

## Molecular dating analysis

To assess how long *Oo* has been circulating within the USA, we generated time-structured phylogenies using BEAST. For these analyses, we focused on 4 subsets of our data, each with a lack of apparent recombination and with evidence for temporal signal as evaluated through comparison of root-to-tip genetic divergence with sampling date (S5 Fig): a mitochondrial analysis that included all 82 strains (Clades I to III; Fig 3) and nuclear analyses focused independently on the 3 most commonly sampled clonal lineages: IID to IIF (Fig 4). Collectively, these 3 clonal lineages accounted for 52% of the *Oo* strains from wild snakes in the USA and are the parental lineages from which another 45% were derived. Across all analyses, substitution rate estimates were very consistent with the reported rates for the amphibian-infecting fungus *Batrachochytrium dendrobatidis* [16] (S6 Fig). The mean estimated rate for the mitochondrial genome of *Oo* was $9.5 \times 10^{-7}$ (95% highest posterior density [HPD]: $3.6 \times 10^{-7}$–$1.6 \times 10^{-6}$) substitutions per site per year using the strict clock model. We do not report results for the relaxed clock with the mitochondrial dataset because we did not observe consistent convergence with this model. The mean estimated nuclear substitution rates for *Oo* were $1.9 \times 10^{-7}$–$1.1 \times 10^{-6}$, including all 3 clonal lineages and both the strict and relaxed clock models (see S4 Table for details).

Based on the mitochondrial analysis, we estimated that Clades I and II diverged approximately 2,000 years ago (mean: 110 CE, 95% HPD: 1362 BCE to 1267 CE), while Clade III diverged from a common ancestor of Clades I and II approximately 11,000 years ago (mean: 9147 BCE, 95% HPD: 17,863 to 2560 BCE) (Fig 3). In contrast, we estimated that the vast majority (74/75) of the Clade II strains last shared a common ancestor approximately 160 years ago (mean: 1860 CE, 95% HPD: 1731 to 1958 CE), while the time of the most recent common ancestor (tMRCA) of Clade I likely occurred approximately 250 years ago (mean: 1770 CE, 95% HPD: 1553 to 1938 CE). Within Clade II, strain NWHC 44736–75 was an outlier. The mitochondrial genome from this strain was estimated to have diverged from the rest of Clade II approximately 1,100 years ago (tMRCA mean: 914 CE, 95% HPD: 22 to 1573 CE).

Across both the strict and relaxed clock nuclear analyses, all of the strains from each individual Clade II clonal lineage were estimated to have most likely shared a common ancestor within the last century (Fig 4 and S5 Table). For lineages IID and IIF, tMRCA estimates were

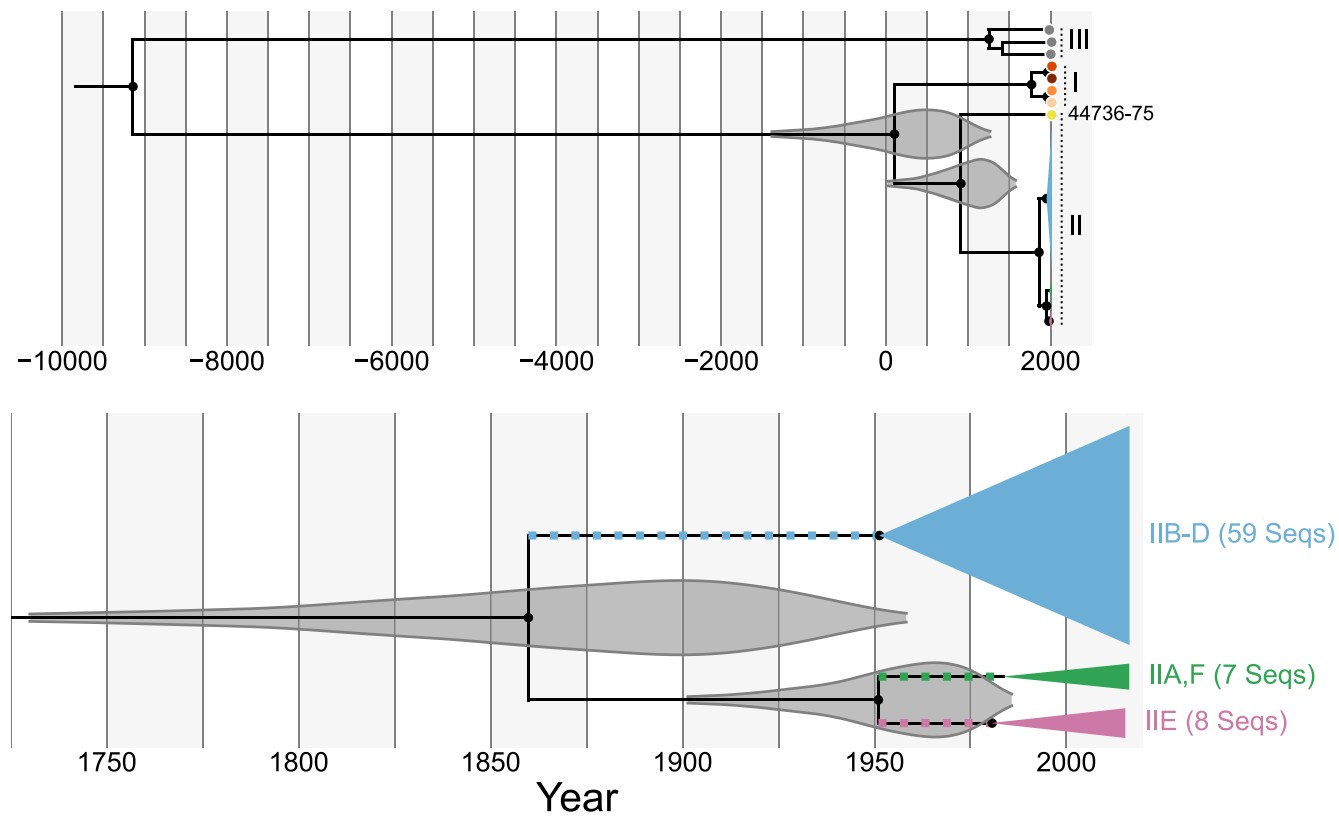

**Fig 3. All 79 strains of *Ophidiomyces ophiodiicola* from Clades I and II likely shared a common ancestor <4,000 years ago.** Time-structured Bayesian phylogeny based on an alignment of 50,624 base pairs of the mitochondrial genome and including all 82 *Oo* strains (see S1C Fig for an uncollapsed, maximum-likelihood version of the same phylogeny). The top panel shows the entire tree, along with the posterior probability distributions (95% HPD) for the tMRCAs of Clades I-II and Clade II (gray). The bottom panel shows just a portion of Clade II and the posterior probability distributions (95% HPD) for the tMRCAs of clonal lineages IID–IIF and IID–IIE. Colored squares indicate the branches on which we infer that clonal lineages IID–IIF were first introduced to North America. Nodes represent mean age estimates. Most of the Clade II strains have been collapsed to save space, and the colors and clade/clonal lineage names are identical to those used in Fig 1. Filled black circles indicate nodes with posterior support ≥95. Data underlying this figure can be found in OSF: https://osf.io/fmbh5/. HPD, highest posterior density; tMRCA, time of the most recent common ancestor.

even more recent when considering only the strains isolated from wild snakes. Mean tMRCA estimates for the isolates from wild snakes from each lineage were 1989/2002, 1985/2007, and 2001/2001 CE for lineages IID, IIE, and IIF, respectively (strict/relaxed clock models), and the oldest date included in any of the 95% HPDs for the wild isolate tMRCAs was 1959 CE (IIE, strict clock analysis). Notably, model testing using both path sampling and stepping stone sampling marginal likelihood estimation approaches found the relaxed clock model to be the best fit for all 3 clonal lineages (S4 Table), and tMRCA estimates for lineages IID and IIE were considerably more recent with the relaxed clock, while the tMRCA estimates for IIF were almost identical with the strict and relaxed models (S5 Table).

Our phylogenetic analysis of these individual lineages also revealed several well-supported clades within these lineages, most of which were geographically restricted within our sample set (Fig 4). Within lineage IIE, for example, our dataset included 3 pairs of strains, each obtained from wild snakes collected from the same state (New Jersey, Ohio, and Wisconsin). Each pair was phylogenetically closely related and 2 formed well-supported sublineages within the phylogeny. However, each isolate was distinct from its closest sampled relative, with mean

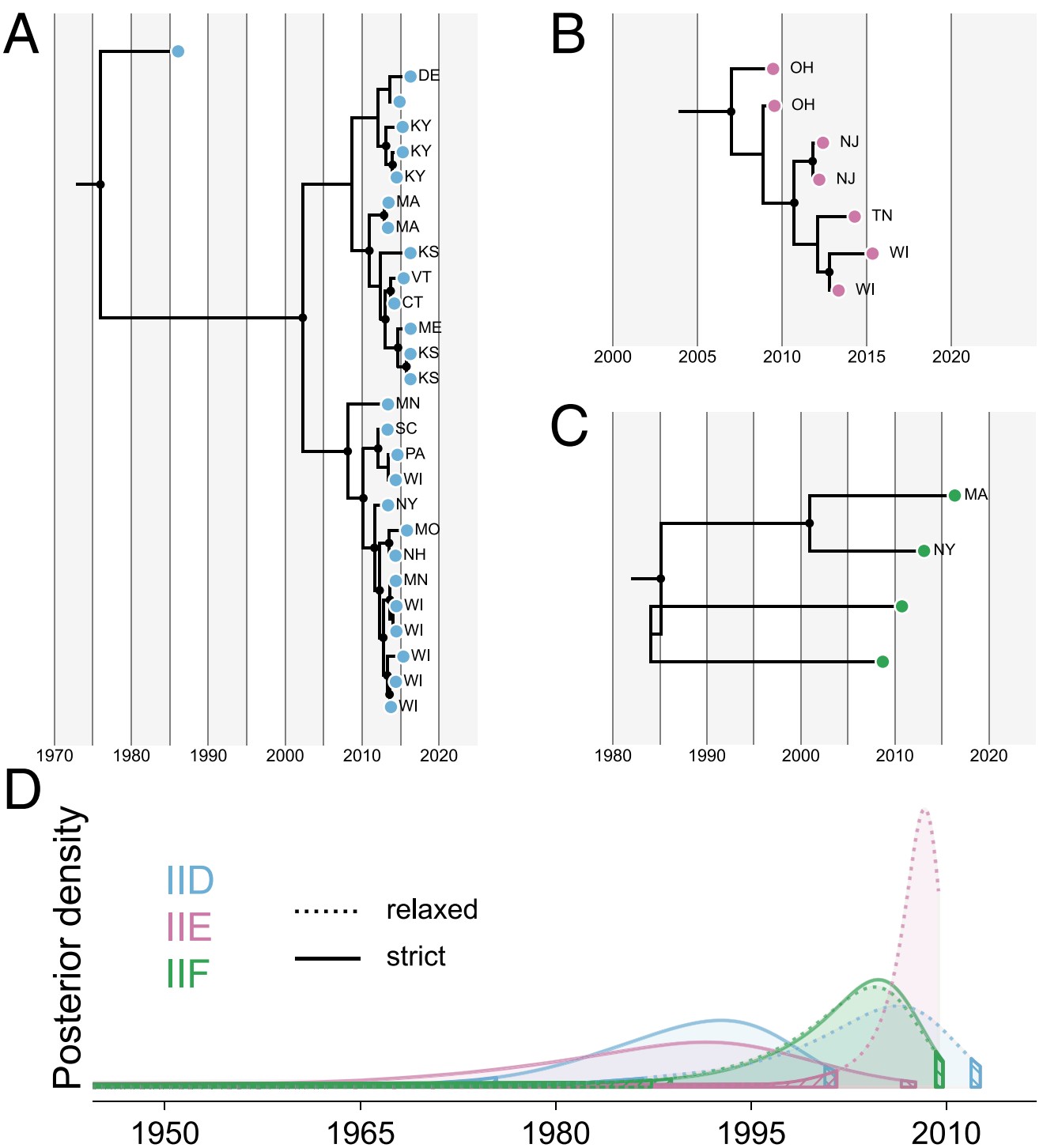

**Fig 4. Strains of *Ophidiomyces ophiodiicola* within the 3 most common clonal lineages on wild snakes in North America likely shared a common ancestor within the last approximately 50–100 years.** (A)–(C) show relaxed clock, time-structured Bayesian phylogenies for Clades IID, IIE, and IIF, respectively. Each phylogeny was built using an alignment of 19,562,499–19,658,811 base pairs within the nuclear genome and nodes represent mean age estimates for each node. Filled black circles indicate nodes with posterior support ≥95. For each strain collected from a wild snake, the state within the USA is indicated using 2-letter abbreviations (strains from captive snakes are not labeled). (D) Posterior probability distributions for the tMRCAs of clonal lineages IID (blue), IIE (pink), and IIF (green). Estimates from analyses using strict and relaxed clock models are shown with solid and dotted lines, respectively. Only strains isolated from wild snakes were considered in the calculation of tMRCAs. Hatched regions represent the tails of each distribution that fall outside of the 95% HPD. Data underlying this figure can be found in OSF: https://osf.io/fmbh5/. HPD, highest posterior density; tMRCA, time of the most recent common ancestor.

estimated divergence times of approximately 1 to 5 years prior to sampling (based on the relaxed clock analysis).

## Discussion

Phylogenetic and population genetic analyses based on whole-genome sequence data can help elucidate the population histories of fungal pathogens and the temporal dynamics of disease emergence. By applying these methods to the causative agent of SFD in the USA, we uncovered a complex history that includes recent expansions of multiple *Oo* clonal lineages as well as recombination between those lineages within a timeframe that is most consistent with the novel pathogen hypothesis [10].

Similar to previous studies that had examined only a small number of loci [26,27], we identified 3 major clades among all 82 sampled *Oo* strains, but all strains originating from wild snakes in the USA belonged to a single clade (Clade II). Within this North American clade, we initially observed what appeared to be high genetic diversity as evidenced by the presence of many long phylogenetic branches (Figs 1C and S1A). However, after accounting for recombination, we identified just 4 clonal lineages within which we saw no evidence of recent recombination, although we did observe evidence of potential historical recombination between these clonal lineages (as indicated by some discordance between nuclear and mitochondrial phylogenies, as well as the presence of moderate levels of homoplasy on the branches separating these lineages). Most remaining North American strains not belonging to these 4 clonal lineages represented recent hybrids between 2 to 3 of those lineages (Fig 2). These findings highlight the importance of accounting for recombination when assessing the evolutionary history of fungal pathogen populations.

We identified 2 mating type loci (*MAT1-1* and *MAT1-2*) among *Oo* strains, consistent with other filamentous ascomycete fungi [37], and both idiomorphs were sampled from wild snakes in the USA at relatively similar frequencies (45% and 55%, respectively). Notably, each clonal lineage within Clade II was characterized by a single mating type and all recombinants involved parent lineages with complementary mating types (i.e., 1 parent lineage possessing *MAT1-1* locus and 1 possessing *MAT1-2* locus). Although clonal expansion was observed among recombinant strains in 2 intensely sampled snake populations (IIB and IIC), most hybrid lineages were sampled only once and had different proportions of their genomes belonging to each parent lineage, indicating unique hybridization events. Furthermore, some strains appeared to be hybrids between more than 2 clonal lineages (Fig 2). Altogether, these patterns indicate that hybridization between *Oo* strains has occurred frequently in the USA as a result of sexual reproduction.

In contrast to the endemic pathogen hypothesis, we estimated that all of the North American lineages of *Oo* shared a common ancestor relatively recently. Assuming that there was only a single introduction of *Oo* to the USA, we would predict that this introduction occurred somewhere along the branch separating the MRCA of Clade II with the MRCA of Clades I and II. We estimated that this occurred approximately 500 to 3,400 years ago according to our mitochondrial phylogeny (Fig 3). The large disparity (4 orders of magnitude) between the dates associated with snake movements across Beringia and the genetic divergence of the USA *Oo* lineages makes it unlikely that the fungus was introduced to North America from elsewhere via natural snake movements.

Although the mitochondrial MRCA of the North American clade (Clade II) is estimated to have occurred long before reported cases of SFD (Fig 3), it is unlikely that this MRCA provides an accurate estimate for the arrival of *Oo* in the USA. The lack of intermediate strains between the main 3 mitochondrial genotypes observed in the USA (blue, green, and pink in Fig 1), the

absence of nonrecombinant intermediates between the nuclear clonal lineages (Fig 2), and the potentially complex history of recombination involving unsampled lineages for strain NWHC 44736–75 all indicate that these *Oo* lineages did not evolve within North America.

Instead, it is more likely that there have been multiple introductions of *Oo* to North America. In fact, the presence of 2 mating types indicates that more than 1 distinct lineage of *Oo* has been introduced to the USA, and the lack of genetic intermediates suggests that each of the nonrecombinant clonal lineages (IIA, IID to IIF), as well as strain NWHC 44736–75, likely represent distinct introductions. Under such a scenario, lineages of *Oo* in the USA would have diverged from a common ancestor that existed outside of North America, and therefore, the MRCA of Clade II would provide an overestimate (i.e., longer ago than is likely realistic) for when the fungus arrived on the continent.

Assuming all 3 of the primary clonal lineages in the USA (IID to IIF) represent separate introduction events, our analyses indicate that these introductions likely occurred within the last few hundred years. When excluding strains originating from captive snakes (which were often basal or divergent from strains of the same clonal lineage that were isolated from wild snakes [Fig 4]), all of the mean tMRCA estimates for these lineages were between 1985 and 2007 (Fig 4), which is shortly prior to the reported emergence of SFD in wild North American snakes. These tMRCAs likely somewhat underestimate the timing of the initial introductions of these lineages, however, as they only represent the timing of the common ancestors of the sampled strains, and our per lineage sample sizes are relatively small, especially for clonal lineages IIE (*n* = 7 nonrecombinant strains from wild snakes) and IIF (*n* = 2). Instead, these should be considered lower bound estimates for the initial introductions of each lineage, while the tMRCAs for IIE to IIF and IID to IIF can provide upper bounds for these estimates. Taken together, our molecular clock analyses indicate that these 3 lineages were likely introduced to North America between 1731 and 2012 for IID and between 1902 and 2009 for IIE and IIF (Fig 3 and S5 Table). These ranges are also consistent with a published survey of museum specimens, which demonstrated that at least 1 lineage of *Oo* was present in the USA as early as 1945 [31].

However, the timing of the tMRCA estimates for the individual clonal lineages (1985 to 2007) likely does reflect quite recent population expansions within the USA. During an invasion, exotic fungal pathogens often exhibit a lag between the introduction, establishment, and expansion phases [38]. Thus, *Oo* may have resided in the USA for decades before widespread expansion occurred (perhaps facilitated by additional spillover events due to capture and movement of infected wild snakes for the pet trade). Older detections of *Oo* on museum specimens collected from the eastern USA could also represent strains that failed to establish or spread after release.

Tracing *Oo* expansion in the USA is difficult due to a lack of historical isolates and frequent recombination between lineages. However, while some clonal lineages (e.g., lineages IID and IIE) were quite widespread throughout the sampled distribution in the USA, other lineages (i.e., IIA and IIF) and hybrids between certain lineages were more geographically restricted (S3 Fig). Of note, many strains sampled from the Atlantic Coastal Plain were hybrids. This was particularly evident in the Gulf Coastal Plain where all strains were unique recombinants, some involving up to 3 parent lineages. Such strain diversity may indicate a longer history of contact between the lineages in this region and imply that the initial introduction of at least some *Oo* lineages occurred in the southern USA. Hybridization between lineages is expected to occur more frequently in regions with high prevalence of multiple *Oo* lineages (i.e., more opportunities for coinfection with strains from distinct lineages). Relatively high prevalence (44% to 78%) has been reported in some snake populations [39–42], which demonstrates the potential for repeated exposure and coinfection. Factors such as host behavior, host pathogen

loads, and environmental parameters, all of which likely vary based on geography, could influence transmission rates and prevalence, resulting in different frequencies of recombination among *Oo* strains.

Despite the broad distributions of several clonal lineages, we did not see evidence for recent long-distance dispersal (S4 Fig). Rather, we observed patterns of genetic divergence within clonal lineages that are consistent with fine-scale population structure (Fig 4). More intensive surveys of *Oo* may reveal spatiotemporal patterns that better elucidate the history of introduction, expansion, and recombination in North America.

In contrast to the recent introductions and subsequent recombination of several clonal lineages in the USA, we identified a single mating type (*MAT1-2*) and no evidence of recombination among our small sample set of *Oo* strains isolated from wild European snakes. The European clade (Clade I) also appears to be older than the North American clonal lineages, with all European strains estimated to have shared an MRCA approximately 100 to 500 years ago. Due to the small sample size (4 strains) and geographic coverage (2 countries), it is probable that we captured only a small portion of the genetic diversity that exists in Europe; thus, we may have underestimated the time at which the European lineage started expanding. Despite this, the MRCA of the European clade still predates widespread anthropogenic movement of snakes [43].

Introduction of *Oo* to North America could have occurred through transcontinental movement of snakes in captive collections. Indeed, *Oo* strains recovered from captive snakes in Australia, Europe, and North America represented 3 separate clonal lineages (and hybrids of those lineages) that were also found in wild snakes in the eastern USA, demonstrating that these strains are circulating in captive snake populations across the world. *Oo* has been detected in soil in Malaysia [44], in snakes imported to Russia from Indonesia [45], and in a wild Burmese python (*Python bivittatus*) in Hong Kong [46]. Furthermore, isolates of *Oo* recovered from wild snakes in Taiwan include a representative that belongs to the North American clade (Clade II), as well as a divergent strain most similar to Clade III [27] based on multi-locus sequence typing analysis (S7 Fig). Repeated detections of *Oo* in Southeast Asia, along with the presence of such diverse strains of *Oo* in Taiwan [27], indicate that the fungus may be native to that region. Many fungal pathogens responsible for wildlife epizootics have their origins in Eurasia, including *B. dendrobatidis* [16], *B. salamandrivorans* [47], and *P. destructans* [33]. Most *Oo* detections outside of North America, including those in Eurasia, have occurred through opportunistic sampling; we are unaware of any comprehensive pathogen surveillance or screening efforts targeting *Oo* in other parts of the world. Thus, *Oo* may be much more widely distributed than is currently documented. Therefore, further sampling would be needed to investigate the "Out of Eurasia" hypothesis as the origin of *Oo* in North America. Additional sampling would also help to better understand the global scope of *Oo* transmission between captive and wild snake populations (e.g., whether Clade II lineages have spilled over into wild snake populations in other parts of the world).

Although it seems likely that *Oo* was originally introduced to the USA many decades ago, it is possible that the recent emergence of SFD in the eastern USA is linked to subsequent, more recent introductions of novel lineages, geographic expansion of certain clonal lineages, and/or recombination between lineages. Indeed, the range of age estimates for the MRCAs of several of the clonal lineages sampled in the northeastern and midwestern USA is consistent with increased reports of SFD in those regions ([18,21,48]; Fig 4). Due to unknown clinical histories and outcomes for most of the snakes sampled in this study, we were unable to assess whether different *Oo* strains or the presence of particular genes were associated with more severe disease. Controlled studies tracing the fate of wild snakes infected with different strains of *Oo* or challenge experiments in the laboratory would be necessary to determine how pathogenicity

varies by lineage, which genes (or combination of genes) might be associated with greater virulence, and whether some hybrid strains possess increased virulence and transmissibility. Recent changes in environmental conditions had previously been proposed to explain the potential emergence of SFD under the endemic pathogen hypothesis [18], and these environmental changes may still be influencing prevalence or virulence, despite the relatively recent introduction of *Oo* to the USA. In other words, treating the novel and endemic pathogen hypotheses as competing explanations for SFD emergence may be overly simplistic. It is feasible that *Oo* could have been introduced into the eastern USA decades ago and emerged as an influential pathogen only recently due to changing environmental conditions and host health that caused a shift in host-pathogen ecology.

Further sampling and strain characterization in other geographic areas would be important to identify the risk posed by *Oo* to snakes on a global scale. For example, naïve snake populations may be most vulnerable to adverse impacts following the introduction of *Oo*. Furthermore, novel contact between previously isolated fungal strains can result in gene transfer and recombination events that may lead to rapid evolution and increased virulence [15,49], especially when the strains have complementary mating types [50]. For example, if additional sampling supports the presence of a single clonal lineage and mating type (*MAT1-2*) in European snake populations, then those populations could be particularly vulnerable to the introduction of additional strains, especially those possessing the *MAT1-1* locus.

Understanding the mechanisms behind disease emergence often starts with elucidating the history of the causative pathogen. Our data demonstrate recent expansion of *Oo* in North America, likely involving a complex history of multiple strain introductions and recombination that has resulted in a diverse strain pool. While additional work is necessary to determine the exact distribution and origin of *Oo*, differences in strain virulence, and the role of environmental factors in SFD outbreaks, our findings provide critical information for proactive management. Specifically, our data highlight that increased vigilance is warranted to prevent further spread of *Oo* and the introduction of novel strains to new areas where snake populations may be at particular risk.

## Materials and methods

### *Ophidiomyces ophiodiicola* isolates

Eighty-two strains of *Oo* (as determined by internal transcribed spacer region sequence data [51]) were included in this study. All of these strains were obtained from publicly accessible culture collections, isolated at the US Geological Survey—National Wildlife Health Center, or shared by other diagnostic laboratories. These strains represented 65 isolates from wild snakes in the eastern USA, 4 isolates from wild snakes in Europe (*n* = 3 from the UK; *n* = 1 from Czech Republic), and 13 isolates from captive snakes on 3 continents (North America, Europe, and Australia). Strains from wild snakes were collected during 2009 to 2016; those from captive snakes were collected over a broader range of time: 1985 to 2015. Strains originated from at least 33 snake species and subspecies, representing 18 genera and 6 families of snakes. The list of all strains used in this study, along with associated metadata, are presented in S2 Table.

### Whole-genome sequencing and annotation

To obtain DNA for whole-genome sequencing, cultures of *Oo* were grown in 20 mL of Sabouraud dextrose broth at 24°C for 9 to 15 days. Fungal biomass from each culture was flash-frozen in liquid nitrogen, lyophilized, pulverized using a pestle and glass beads, resuspended in 700 μL of LETS buffer (20 mM EDTA [pH 8.0], 0.5% sodium dodecyl sulfate, 10 mM Tris–HCl [pH 8.0], 0.1 M LiCl), and DNA was extracted using the phenol-chloroform method.

Library preparation and whole-genome sequencing were performed by the University of Wisconsin-Madison Biotechnology Center (Madison, Wisconsin). Seven strains (NWHC 22687–1, NWHC 23942–1, NWHC 24266–2, NWHC 45692–2, CBS 122913 [type isolate of *Oo*], UAMH 10296, and UAMH 9985) were sequenced on an Illumina MiSeq system (2 × 250 bp read lengths). The remaining 75 strains were sequenced on an Illumina HiSeq 2500 sequencing system using a high-output kit and 2 × 125 bp read lengths.

Nuclear and mitochondrial de novo assemblies were generated using the automatic assembly for the fungi (AAFTF) pipeline [52]. Briefly, the AAFTF pipeline ran the following steps: the Illumina data were cleaned (i.e., removal of adapters, artifacts, contaminants, etc.) using BBDuk from the BBMAP v38.92 package [53], the mitochondrial genome was assembled using NOVOplasty v4.2 [54], the nuclear genome was assembled with SPAdes v3.10.1 [55], and the resulting nuclear assembly was error corrected with 3 polishing rounds using Pilon v1.24 [55,56]. Assembled genomes were deposited in the National Center for Biotechnology Information (NCBI) database under BioProject PRJNA780910 (see S2 Table).

Genome annotation was completed using funannotate v1.5.2 [57]. A library of repetitive sequences found in *Oo* was generated for the type strain (CBS 122913) using the funannotate wrapper for RepeatModeler v1.0.11 [58] and RepeatMasker v4.0.7 [59]. The resulting library was used to soft-mask all of the de novo assemblies in this study using the "funannotate mask" command. Subsequently, de novo gene prediction training parameters for AUGUSTUS v3.2.1 [60] were generated from strain CBS 122913 by utilizing the automated training pipeline in "funannotate predict." Briefly, BUSCO [34] was used to generate initial gene predictions used for training AUGUSTUS. The AUGUSTUS training parameters were used on all genomes in this study. The "funannotate predict" pipeline also utilized gene predictions from GeneMark v4.32 [61] to compute consensus gene model predictions using EvidenceModeler [62].

## Mating type

To identify the mating type idiomorph in whole-genome sequence datasets, we aligned the Illumina reads for each strain to the CBS 122913 genome assembly using minimap2 v2.12 [63]. We then calculated the ratio of read mapping coverage at the *MAT1-1* locus (Gen Bank: JALAZP010000006.1:201703–202676) compared to a region outside of the mating type idiomorph (GenBank:JALAZP010000006.1:196703–197676) using BEDTools [64]. To identify *MAT1-2*, we aligned the conserved flanking regions of *MAT1-1* from strain CBS 122913 with those of a strain in which *MAT1-1* was not detected and identified predicted genes between those flanking regions. To further investigate which *MAT1* loci were present in a larger sample set from Europe, we screened 9 additional strains of *Oo* isolated from wild snakes in the UK for which whole-genome sequencing was not performed for the presence of *MAT1-1* and *MAT1-2* using 2 newly designed PCR assays (see S1 File and S2 Table).

## Phylogenetic analysis

We used Proteinortho v6 [65] to identify putatively orthologous nuclear protein-coding genes across all 82 *Oo* strains using amino acid sequences as input. We individually aligned amino acid sequences for each of the 5,811 single copy genes present in all 82 isolates using MAFFT v7.487 with default settings [66], and then concatenated all of these aligned proteins, which resulted in a single amino acid-level, genome-wide alignment that consisted of 3,311,400 positions. Using this alignment, we generated a maximum-likelihood phylogeny using RAxML v8.2.12 [67]. We used RAxML's automatic protein model assignment algorithm (-m PROT-GAMMAAUTO), which selected the JTT model as the best fit, along with 100 rapid bootstrapping iterations followed by a maximum-likelihood search (-f a -N 100).

We identified SNPs using NASP v1.2.0 [68] with either Bowtie2 v2.4.2 [69] (mitochondrial genome) or BWA-mem v0.7.7 [70] (nuclear genome) to align reads (default settings) and with GATK v3.4-46-gbc02625 [71,72] to call variants (-T UnifiedGenotyper -dt NONE -glm BOTH -out_mode EMIT_ALL_CONFIDENT_SITES -baq RECALCULATE -stand_call_conf 100 -ploidy 1). MUMmer v3.23 [73] was used to identify duplicated regions within our reference sequences (nucmer, find_duplicates). To avoid potentially erroneous genotype calls, we ignored any SNPs present within duplicated regions. For calling SNPs within the nuclear genome, we considered the 79 *Oo* strains from Clades I and II and utilized 38 scaffolds previously assembled from the MYCO-ARIZ AN0400001 strain of *Oo* as a reference (GenBank: MWKM01000001.1—MWKM01000021.1, MWKM01000023.1—MWKM01000039.1). For calling SNPs within the mitochondrial genome, we considered all 82 *Oo* strains and used a custom de novo assembly for the type strain of *Oo*, CBS 122913 (GenBank: CM040671.1), as a reference. We chose to use a custom reference for the mitochondrial genome because the mitochondrial scaffold for MYCO-ARIZ AN0400001 (GenBank: MWKM01000023.1) was incomplete, containing 15 large gaps in contiguous sequence. For both the nuclear and mitochondrial genomes, we utilized all reference positions at which a high confidence call was made for ≥80% of the included strains. Prior to running NASP, we removed sequencing adapters and trimmed low quality bases using BBDuk (last modified 2019-01-23) (bbduk.sh ktrim = r tpe tbo k = 23 mink = 8 hdist = 1 hdist2 = 1 ftm = 5 ref = ~/bbmap/resources/adapters.fa; bbduk.sh qtrim = rl trimq = 15 maq = 20 ref = ~/bbmap/resources/phix174_ill.ref.fa.gz k = 31 hdist = 1 minlen = 50) [53]. We then generated maximum-likelihood phylogenies using RAxML-NG v0.5.1b [74] with the GTR+G model, 10 randomized parsimony starting trees, and 100 bootstrap replicates. For the mitochondrial SNP alignment, we also constructed a median-joining haplotype network [75] using PopART [76].

## Recombination analyses

We used PAUP* v4.0a (build 168) to build parsimony-based phylogenies and to calculate consistency/homoplasy indexes. Trees were constructed using random stepwise addition with 100 replicates and tree-bisection-reconnection with a reconnection limit of 8. PhiPack [77] was used to examine recombination using the Phi test [35]. Analyses were performed individually for each reference scaffold (using all positions with high confidence calls in ≥80% of the 75 strains from Clade II) with a window size of 50,000 bases and a step size of 25,000 bases.

To generate a co-ancestry matrix for the 79 strains from Clades I and II, we used ChromoPainter within FineSTRUCTURE v4.1.1 [36]. We used the default parameters implemented via the "automatic" mode, with ploidy set to 1 and a fixed recombination rate of 0.000001 Morgans. We then conducted a PCA using the "chunkcounts" file generated by ChromoPainter along with the *mypca* function contained in the FineSTRUCTURE R library [78].

The capture locations for most snakes were available only to the county level. Thus, we used county centroids (or state centroids when state was the only locality information available) as coordinates to represent origin locations for *Oo* strains. Distances between strain collection localities were calculated with the Haversine formula using the geosphere package [79] in R [80].

## Molecular dating analysis

To estimate dates of divergence for the major *Oo* clades, we used BEAST v1.10.5 with BEAGLE v.3.2.0 [81]. For the mitochondrial genome, we conducted a single analysis including all 82 strains and all reference positions at which a genotype was called in ≥80% of the strains (50,624 bp, 817 variable sites). Although we observed some discordance between nuclear and

mitochondrial phylogenies within Clade II, we did not observe any discordance in clade membership, and therefore analysis of the mitochondrial genome could be used to date the divergence between clades. Because of a history of recombination in the nuclear genome, we ran separate analyses for each of the 3 primary clonal lineages in Clade II (IID to IIF) using all reference positions at which a genotype was called for 100% of the included strains (19,562,499 to 19,658,811 bp, 324 to 1,635 variable sites). For the nuclear datasets, we used ClonalFrameML [82] to identify and remove any regions with evidence of recombination. This analysis only flagged 16, 41, and 0 bp as potentially recombinant for clonal lineages IID, IIE, and IIF, respectively.

All 4 analyses used the GTR substitution model including invariant sites and a discrete gamma distribution of rates among sites with 4 categories and a coalescent tree model assuming constant population size over time [83] with a lognormal prior distribution ($\mu = 0$, $\sigma = 1$, offset = 0). For each dataset, we compared 2 clock models: a strict clock and an uncorrelated relaxed clock model with a lognormal distribution, both with a continuous-time Markov chain rate reference prior [84]. Each analysis was run with a chain length of $\geq$500 M and parameters were logged 10,000 times per run, at regular intervals. Each dataset and model combination was run at least twice with different random seeds to ensure consistent convergence. We also tested additional tree models (e.g., Bayesian Skyline and Bayesian SkyGrid); however, none of our datasets reached convergence with these more complex tree priors.

## Supporting information

**S1 Fig. All North American *Ophidiomyces ophiodiicola* (*Oo*) strains isolated from wild snakes belong to a distinct phylogenetic clade.** (A) Maximum-likelihood phylogeny including all 82 *Oo* strains and based on an amino acid-level alignment consisting of 5,811 nuclear proteins and 3,311,400 positions. (B) Boxplot showing the mean, pairwise percent divergence between Clade III and Clades I and II across the 5,811 proteins included in the alignment for (A). The red dashed lines indicate the values for the 2 protein coding genes that have been used previously in phylogenetic studies of *Oo*: actin (*ACT*) and translation elongation factor 2α (*TEF*). The limits of the box correspond to the first and third quartiles, the black line inside the box corresponds to the median, and the whiskers extend to points that lie within 1.5 interquartile ranges of the first and third quartiles. (C) Maximum-likelihood phylogeny including all 82 *Oo* strains and based on a reference-based, nucleotide-level mitochondrial alignment (50,624 positions). In both (A) and (C), tip shapes indicate whether the infected snake was in captivity (triangles) or wild (circles). Filled black and white circles indicate nodes with bootstrap support $\geq$90 and $\geq$70, respectively. Gray vertical lines and labels indicate the 3 primary clades. Data underlying this figure can be found in OSF: https://osf.io/fmbh5/.
(PDF)

**S2 Fig. Evidence for recombination within Clade II throughout the *Ophidiomyces ophiodiicola* genome.** Log-transformed Phi test *p*-values calculated using PhiPack with a window size of 50,000 bases and a step size of 25,000 bases (1). Black horizontal dashed line indicates a *p*-value of 0.05 with Bonferroni correction for 749 different tests. Red vertical dashed lines indicate scaffold boundaries within the reference assembly. Data underlying this figure can be found in OSF: https://osf.io/fmbh5/.
(PDF)

**S3 Fig. Geographic distribution of *Ophidiomyces ophiodiicola* strains isolated from wild snakes in the eastern USA.** Solid circles represent nonrecombinant clonal lineages. Divided circles depict recombinant strains (color combinations qualitatively indicate genetic signatures

of the various lineages present in those recombinants [see Fig 2]). For strain NWHC 44736–75, a portion of the circle is gray to indicate an unsampled parent lineage. Circle sizes denote the number of strains when multiple strains were collected in proximity to one another. The edge of the Atlantic and Gulf Coastal Plain region is shown with a dashed line. Data underlying this figure can be found in OSF: https://osf.io/fmbh5/. The base map used to generate this figure is from Natural Earth (https://www.naturalearthdata.com) and available through GitHub (https://github.com/nvkelso/natural-earth-vector/blob/master/geojson/ne_50m_admin_1_states_provinces_lakes.geojson).
(PDF)

**S4 Fig. Within clonal lineages of *Ophidiomyces ophiodiicola* (*Oo*), there is no evidence of recent, long-distance dispersal.** Great-circle distance (x-axis) versus pairwise nucleotide divergence (y-axis) for each pair of strains that were isolated from wild snakes and belonged to the same clonal lineage within Clade II. Each point represents a pair of strains, and the colors indicate the clonal lineages. Genetic divergence was calculated using 116,322 positions in the nuclear genome that were variable among *Oo* strains from Clade II. Data underlying this figure can be found in OSF: https://osf.io/fmbh5/.
(PDF)

**S5 Fig. Root-to-tip plots demonstrate the presence of molecular clock signals within both the mitochondrial and nuclear genomes of *Ophidiomyces ophiodiicola*.** Root-to-tip divergence (y-axis) was calculated using TempEst (2) v1.5.3 using maximum-likelihood phylogenies, and the colored numbers inside each plot represent Pearson correlation coefficients. The mitochondrial tree (left) was rooted using the residual mean squared best-fitting root function. The nuclear trees for the Clade II clonal lineages (right) were rooted using the correlation best-fitting root function. A positive slope indicates the presence of a molecular clock signal. Data underlying this figure can be found in OSF: https://osf.io/fmbh5/.
(PDF)

**S6 Fig. Substitution rate estimates for *Ophidiomyces ophiodiicola* (*Oo*) are very similar to published estimates for *Batrachochytrium dendrobatidis* (*Bd*).** Colored distributions represent the posterior probability distributions from BEAST for the *Oo* substitution rate in number of substitutions per site per year. There is a single rate estimate for the mitochondrial genome based on a strict clock model (top, red) and 6 rate estimates for the nuclear genome, 2 for each of the 3 primary clonal lineages based on the strict (solid line) and relaxed (dotted line) clock models. Hatched regions represent the tails of each distribution that fall outside of the 95% highest posterior density. Gray boxes indicate published rate estimates for *Batrachochytrium dendrobatidis* (*Bd*) (3), with the edges of the boxes indicating the boundaries of the 95% highest posterior densities and the dashed lines representing the mean estimates. Data underlying this figure can be found in OSF: https://osf.io/fmbh5/.
(PDF)

**S7 Fig. Phylogenetic tree of *Ophidiomyces ophiodiicola* strains, including all strains from Clades I and III, nonrecombinant strains from Clade II, and strains isolated from wild snakes in Taiwan, based on 3 concatenated loci (internal transcribed spacer region, actin, and translation elongation factor 2α).** Both Bayesian and maximum-likelihood analyses produced trees with identical topologies (consensus tree from Bayesian analysis is shown). Posterior probabilities/bootstrap support values are shown at each node. Note that 2 strains recovered from wild snakes in Taiwan (shown in yellow) are most closely related to Clade III and 2 strains reside within Clade II (i.e., North American clade). Data underlying this figure

can be found in OSF: https://osf.io/fmbh5/.
(PDF)

**S1 Table. Per strain summary of high-throughput whole-genome sequence data and resulting de novo genome assemblies.**
(XLSX)

**S2 Table. Metadata for strains of *Ophidiomyces ophiodiicola* used in this study.**
(XLSX)

**S3 Table. Results of PAUP* analysis for various subsets of the *Ophidiomyces ophiodiicola* strains sequenced in this study.**
(XLSX)

**S4 Table. Summary of BEAST model testing and substitution rate estimates.**
(XLSX)

**S5 Table. Summaries of the posterior probability distributions from BEAST for the date estimates at which the most recent common ancestors of various *Ophidiomyces ophiodiicola* clades existed.**
(XLSX)

**S1 File. Supporting results and methods.**
(PDF)

## Acknowledgments

We thank the numerous state and federal biologists, contractors, wildlife health staff, scientists, and volunteers, including personnel at the US Geological Survey—National Wildlife Health Center, that participated in collecting and compiling metadata, as well as collecting and processing samples that led to the isolation of *Ophidiomyces ophiodiicola* strains from the USA. We also thank the Garden Wildlife Health project (www.gardenwildlifehealth.org, a wildlife disease surveillance program coordinated by the Institute of Zoology, UK), Angela Winnett, Dr. Iain Barr, University of East Anglia, UK, Dr. Dave Leech, British Trust for Ornithology, UK, and Dr. Vojtech Baláž, University of Veterinary and Sciences Brno who similarly assisted with collection of metadata and samples in Europe. We are grateful to the various laboratories that preserved and shared strains that were previously published on, including the Cornell University College of Veterinary Medicine (Dr. Krysten Schuler), the Southeastern Cooperative Wildlife Disease Study, and the University of Florida College of Veterinary Medicine (Dr. Jim Wellehan).

Any use of trade, firm, or product names is for descriptive purposes only and does not imply endorsement by the US Government.

## Author Contributions

**Conceptualization:** Jason T. Ladner, Daniel A. Grear, Jeffrey M. Lorch.

**Data curation:** Jason T. Ladner, Jonathan M. Palmer, Cassandra L. Ettinger, Jason E. Stajich, Terence M. Farrell, Brad M. Glorioso, Becki Lawson, Steven J. Price, Anne G. Stengle, Jeffrey M. Lorch.

**Formal analysis:** Jason T. Ladner, Jonathan M. Palmer, Cassandra L. Ettinger, Jason E. Stajich, Jeffrey M. Lorch.

**Funding acquisition:** Jeffrey M. Lorch.

**Investigation:** Jason T. Ladner, Jonathan M. Palmer, Cassandra L. Ettinger, Jason E. Stajich, Terence M. Farrell, Brad M. Glorioso, Becki Lawson, Steven J. Price, Anne G. Stengle, Jeffrey M. Lorch.

**Methodology:** Jason T. Ladner, Jonathan M. Palmer, Cassandra L. Ettinger, Jason E. Stajich, Daniel A. Grear, Jeffrey M. Lorch.

**Project administration:** Jason T. Ladner, Jeffrey M. Lorch.

**Resources:** Jason T. Ladner, Daniel A. Grear, Jeffrey M. Lorch.

**Software:** Jason T. Ladner, Jonathan M. Palmer, Cassandra L. Ettinger, Jason E. Stajich.

**Supervision:** Jason E. Stajich, Jeffrey M. Lorch.

**Validation:** Jason T. Ladner, Jason E. Stajich, Jeffrey M. Lorch.

**Visualization:** Jason T. Ladner, Jeffrey M. Lorch.

**Writing – original draft:** Jason T. Ladner, Jeffrey M. Lorch.

**Writing – review & editing:** Jason T. Ladner, Jonathan M. Palmer, Cassandra L. Ettinger, Jason E. Stajich, Terence M. Farrell, Brad M. Glorioso, Becki Lawson, Steven J. Price, Anne G. Stengle, Daniel A. Grear, Jeffrey M. Lorch.

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
