## [Editor Report · Decision Letter 0]

3 Feb 2022

Dear Dr Lorch, 

Thank you for submitting your manuscript entitled "Population genetic analysis of Ophidiomyces ophidiicola, the causative agent of snake fungal disease, suggests recent introductions to the USA" for consideration as a Research Article by PLOS Biology.

Your manuscript has now been evaluated by the PLOS Biology editorial staff and I am writing to let you know that we would like to send your submission out for external peer review. Please accept my sincere apologies for the delay in communicating this decision, which was due to a backlog of submissions after the office closure over Christmas.

Despite the delay, we have unfortunately not managed to obtain expert advice from an Academic Editor regarding the advance represented by your paper, but have decided, in the interests of time, to proceed with peer review without such advice.

I should warn you that while we appreciate the conservation angle of your paper, we're less clear about the overall conceptual advance of the study, and we'll be relying on the reviewers' assessment to judge this aspect. 

Before we can send your manuscript to reviewers, we need you to complete your submission by providing the metadata that is required for full assessment. To this end, please login to Editorial Manager where you will find the paper in the 'Submissions Needing Revisions' folder on your homepage. Please click 'Revise Submission' from the Action Links and complete all additional questions in the submission questionnaire.

Once your full submission is complete, your paper will undergo a series of checks in preparation for peer review. Once your manuscript has passed the checks it will be sent out for review. To provide the metadata for your submission, please Login to Editorial Manager (https://www.editorialmanager.com/pbiology) within two working days, i.e. by Feb 04 2022 11:59PM.

If your manuscript has been previously reviewed at another journal, PLOS Biology is willing to work with those reviews in order to avoid re-starting the process. Submission of the previous reviews is entirely optional and our ability to use them effectively will depend on the willingness of the previous journal to confirm the content of the reports and share the reviewer identities. Please note that we reserve the right to invite additional reviewers if we consider that additional/independent reviewers are needed, although we aim to avoid this as far as possible. In our experience, working with previous reviews does save time. 

If you would like to send previous reviewer reports to us, please email me at dummarino@plos.org to let me know, including the name of the previous journal and the manuscript ID the study was given, as well as attaching a point-by-point response to reviewers that details how you have or plan to address the reviewers' concerns. 

Given the disruptions resulting from the ongoing COVID-19 pandemic, please expect some delays in the editorial process. We apologise in advance for any inconvenience caused and will do our best to minimize impact as far as possible.

Kind regards,

Dario

Dario Ummarino, PhD

Senior Editor

PLOS Biology

dummarino@plos.org

---

## [Decision Letter · Decision Letter 1]

7 Apr 2022

Dear Dr Lorch,

Thank you for submitting your manuscript "Population genetic analysis of Ophidiomyces ophidiicola, the causative agent of snake fungal disease, suggests recent introductions to the USA" for consideration as a Research Article at PLOS Biology. Your manuscript has been evaluated by the PLOS Biology editors, an Academic Editor with relevant expertise, and by independent reviewers. Please accept my sincere apologies for the delayed editorial process, which was due to difficulties in securing reviewers with relevant expertise. 

As you will see in the reviews attached below, all reviewers appreciate the importance and accuracy of your work, and raised minor points around methodology, reporting and discussion/interpretation of the results. In light of these reviews, we are pleased to offer you the opportunity to address the comments from the reviewers in a revised version that we anticipate should not take you very long. We will then assess your revised manuscript and your response to the reviewers' comments and we may consult the reviewers and the Academic Editor again. We also request that your please address the following data and other policy-related requests:

1) Title - we suggest minor changes for clarity and appeal: "The population genetics of the causative agent of snake fungal disease suggests recent introductions to the USA".

2) Article type change: We request that your article is converted into a *Short Report*. Short Reports present the results from a limited set of experiments that can generally be summarized in 3-4 figures or fewer. The outcomes should be self contained, rather than fitting within the narrative arc of a larger research project or article. More information about this article type can be found in this Editorial: https://journals.plos.org/plosbiology/article?id=10.1371/journal.pbio.3000248. To meet our format requirements for Short Report articles, please reduce the number of main figures to 4 maximum by either combining some of the existing ones into a single multi-panel figure, or by moving one or more to the supplementary material. Please also ensure that you select the 'Short Report' article type in the online submission form when you submit your revisions. 

3) Data: you may be aware of the PLOS Data Policy, which requires that all data be made available without restriction: http://journals.plos.org/plosbiology/s/data-availability. For more information, please also see this editorial: http://dx.doi.org/10.1371/journal.pbio.1001797

Note that we do not require all raw data. Rather, we ask for all individual quantitative observations that underlie the data summarized in the figures and results of your paper. For an example see here: http://www.plosbiology.org/article/info%3Adoi%2F10.1371%2Fjournal.pbio.1001908#s5

These data can be made available in one of the following forms:

I) Supplementary files (e.g., excel). Please ensure that all data files are uploaded as 'Supporting Information' and are invariably referred to (in the manuscript, figure legends, and the Description field when uploading your files) using the following format verbatim: S1 Data, S2 Data, etc. Multiple panels of a single or even several figures can be included as multiple sheets in one excel file that is saved using exactly the following convention: S1_Data.xlsx (using an underscore).

II) Deposition in a publicly available repository. Please also provide the accession code or a reviewer link so that we may view your data before publication.

Regardless of the method selected, please ensure that you provide the individual numerical values that underlie the summary data displayed in the following figure panels: 1AB, 2ABC, 3, 4, 5 ABCD, S1 ABC, S2, S3, S4, S5, S6.

3.1) IMPORTANT: Please also cite the location of the data clearly in each relevant main and supplementary Fig legend, e.g. “Data underlying this Figure can be found in S1 Data”.

3.2) Please ensure that your Data Statement in the submission system accurately describes where your data can be found. 

We expect to receive your revised manuscript within 1 month.

**IMPORTANT - SUBMITTING YOUR REVISION**

*Resubmission Checklist*

*Published Peer Review*

Sincerely,

Dario

Dario Ummarino, PhD

Senior Editor

PLOS Biology

dummarino@plos.org

REVIEWS:

Reviewer #1: In the paper "Population genetic analysis of Ophidiomyces ophidiicola, the causative agent of snake fungal disease, suggests recent introductions to the USA", the authors sequence 82 strains of the snake fungal disease Oo and perform genomic, population genetic and dating analysis to predict the timing and location of its emergence in the United States.

Overall, it is very nice study, with clear results regarding the population of Oo in north America. Correspondingly, I only have very minor suggestions for improvement. 

Firstly, given the authors have generated a new reference strain, I'd like to see a BUSCO gene completeness report for that (E.g. what percent of single copy orthologs were found?). It would also be nice to give further details about any previous genome assemblies that were available, and why these were deemed unsuitable / why they generated a new one?

Secondly, for variant calling with GATK, I'd like a proper description of the pipeline, including all commands used, and in particular, a description of any hard or soft filters used. The current description of variant calling would not be sufficient to replicate this study.

Finally, I think the tree with all clades i.e. clade III (currently in supplemental) should probably be added to Figure 1 (perhaps as part A, while the other 2 parts become B and C). I'm not certain what the rationale of having it supplemental is, other than it appearing much more distant. For the Figure 1 legend, you should also describe what the branch lengths show / the length key.

Otherwise, this was an enjoyable and informative report on their work.

Reviewer #2: This is an important study on the population genetic history of an emerging fungal pathogen affecting a large and diverse group of wildlife. I think the work warrants close consideration by PLOS Biology. I really only have one significant concern, which is that the distinction between the two hypotheses seems somehow artificial. We understand the range of historical possibilities without arbitrarily designating these as two different hypotheses, although I recognize scientific writers differ in style sometimes to achieve the appearance of adherence to the relatively simplistic concepts of the "hypothetico-deductive method". The point is, I would welcome the authors to describe the historical possibilities with greater nuance and richness rather than adopt the binary either/or of the "novel pathogen hypothesis" vs "endemic pathogen hypothesis", especially since their own findings support a much more complicated pattern of introduction. Also, might it not be the case that the relatively recent detections of Oo are due to environmental change or stress, in which case the "emergence" of this disease should be causally attributed to both recent (post-colonial) introductions AND environmental changes?

I also have a few questions that the study does not address, but which might be considered in the discussion.

- Why is Clade II not now spreading in other parts of the world?

- What mechanisms might be offered to explain recombination, i.e. under what conditions might coinfection of different lineages within Clade II have occurred?

Overall, this is a well-conducted and well-written study. One qualification I would like the authors and editors to note is that I do not consider myself sufficiently qualified to evaluate the specific details of the bioinformatic pipeline.

---

## [Editor Report · Decision Letter 2]

13 May 2022

Dear Dr Lorch,

On behalf of my colleagues and the Academic Editor, Andy Dobson, I am pleased to say that we can in principle accept your Short Reports "The population genetics of the causative agent of snake fungal disease indicate recent introductions to the USA" for publication in PLOS Biology, provided you address any remaining formatting and reporting issues. These will be detailed in an email that will follow this letter and that you will usually receive within 2-3 business days, during which time no action is required from you. Please note that we will not be able to formally accept your manuscript and schedule it for publication until you have completed any requested changes.

PRESS

Sincerely, 

Dario Ummarino, PhD 

Senior Editor 

PLOS Biology

dummarino@plos.org